# Characterization of the Calmodulin/Calmodulin-like Protein (CAM/CML) Family in *Ginkgo biloba*, and the Influence of an Ectopically Expressed *GbCML* Gene (*Gb_30819*) on Seedling and Fruit Development of Transgenic *Arabidopsis*

**DOI:** 10.3390/plants11111506

**Published:** 2022-06-04

**Authors:** Xinxin Zhang, Juan Tian, Sai Li, Yuying Liu, Ting Feng, Yunyun Wang, Yuanjin Li, Xinxin Huang, Dahui Li

**Affiliations:** College of Life Sciences, Anhui Agricultural University, Hefei 230036, China; zhangxinxin@stu.ahau.edu.cn (X.Z.); tianjuan123@stu.ahau.edu.cn (J.T.); lisai@stu.ahau.edu.cn (S.L.); liuyuying@stu.ahau.edu.cn (Y.L.); fengting@stu.ahau.edu.cn (T.F.); wangyunyun@stu.ahau.edu.cn (Y.W.); liyuanjin@stu.ahau.edu.cn (Y.L.); huangxinxin@stu.ahau.edu.cn (X.H.)

**Keywords:** calcium, calmodulin, calmodulin-like protein, gene family, *Ginkgo biloba*, transcriptional expression

## Abstract

Calmodulins (CAMs) and calmodulin-like proteins (CMLs) can participate in the regulation of various physiological processes via sensing and decoding Ca^2+^ signals. To reveal the characteristics of the CAM/CML family in *Ginkgo biloba*, a comprehensive analysis was performed at the genome-wide level. A total of 26 *CAMs/CML*s, consisting of 5 *GbCAM*s and 21 *GbCML*s, was identified on 11 out of 12 chromosomes in *G. biloba*. They displayed a certain degree of multiplicity in their sequences, albeit with conserved EF hands. Collinearity analysis suggested that tandem rather than segmental or whole-genome duplications were likely to play roles in the evolution of the *Ginkgo CAM/CML* family. Furthermore, *GbCAMs/GbCML*s were grouped into higher, lower, and moderate expression in magnitude. The cis-acting regulatory elements involved in phytohormone-responsiveness within *GbCAM/GbCML* promotors may explain their varied expression profiles. The ectopic expression of a *GbCML* gene (*Gb_30819*) in transgenic *Arabidopsis* led to phenotypes with significantly shortened root length and seedling height, and decreased yields of both pods and seeds. Moreover, an electrophoresis mobility shift assay demonstrated the Ca^2+^-binding activity of Gb_30819 in vitro. Altogether, these results contribute to insights into the characteristics of the evolution and expression of *GbCAMs/GbCML*s, as well as evidence for Ca^2+^-CAM/CML pathways functioning within the ancient gymnosperm *G. biloba*.

## 1. Introduction

Intracellular Ca^2+^ signals that emerge in the form of transiently enhanced calcium concentration within the cytosol are elicited by certain stimuli, and coupled with peculiar cellular responses to developmental or environmental factors [1,2]. Therefore, calcium has been considered as an important secondary messenger in various signaling pathways [3]. The functions of Ca^2+^ in a multitude of modulatory aspects generally rely on Ca^2+^-binding proteins, designed as sensors, which play key roles in both decoding and transducing Ca^2+^ signals by activating target proteins within the specific pathways [1,4].

The majority of known Ca^2+^ sensors have been found to have a conserved EF-hand Ca^2+^-binding domain [5]. These EF-hand-containing proteins in plants constitute a superfamily, which are predicted to fulfill the versatile modulation of Ca^2+^ signals, although their functions have not yet been elucidated in most cases [6]. Calmodulin (CAM) is one of the most well-characterized Ca^2+^ sensors in both animals and plants [7]. In contrast to CAMs, which are widely found in eukaryotes, calmodulin-like proteins (CMLs) seem to be plant-specific [3,8,9]. The CAM/CML family in *Arabidopsis thaliana* has been extensively elucidated [8,10]. The *Arabidopsis* genome harbors 7 *AtCAM* and 50 *AtCML* genes, among which there appears to be a great of range in their sequence identities. The 7 CAMs are classified into four types of protein isoforms based on sequence difference in 1–4 amino acids: AtCAM1 and AtCAM4 in type one; AtCAM2, AtCAM3, and AtCAM5 in type two; and AtCAM6 and AtCAM7 in types three and four, respectively [10]. The comparison of AtCAMs/AtCMLs showed that sequence identities among the AtCAMs were more than 90%, whereas those between each of the AtCAMs and AtCMLs were less than 50%, indicating that AtCMLs have undergone highly structural divergence and adopted novel functions [7,10]. Structurally, all 7 AtCAMs contain two pairs of EF hands. Comparatively, although a majority of AtCMLs (31 out of 50) possess two pairs of EF hands, there are 17 AtCMLs with one pair of EF hands, and AtCML12 with three pairs of EF hands, in addition to AtCML1 without identifiable EF hands [7,10]. Despite their variation in numbers of EF hands, there exist highly conserved motifs of the Ca^2+^-binding loops within the EF hands of *Arabidopsis* CAMs/CMLs.

With the completion of genome sequencing in several plant species, the characterization of the *CAM/CML* gene family at the genome-wide level has been carried out in *A. thaliana* [8], rice (*Oryza sativa*) [11], *Fragaria vesca* [12], tomato (*Solanum lycopersicum*) [13], in cabbage (*Brassica rapa* L. ssp. *pekinensis*) [14], grapevine (*Vitis vinifera*) [15], and apple (*Malus domestica*) [16]. Accumulating evidence has demonstrated that CAM/CML family members are involved in both plant development and responses to environmental stress [17]. One of the *Arabidopsis* CAMs (AtCAM7) has been proven to control the photomorphogenesis of seedlings [18], while another member (AtCAM2) functions in pollen germination and, thereafter, pollen tube growth [19]. By mediating Ca^2+^ signals, certain CAMs may coordinate with other signaling pathways—for instance, a crosstalk network with abscisic acid, auxin, or brassinosteroids [20,21,22,23]. One member of the *Arabidopsis* CMLs (AtCML42) could be a regulator for microtubule organization, leading to defect morphology of trichomes in the event of its knockout mutant [24]. Similarly, AtCML24 has an effect on pollen tube growth by modulating the dynamics of the actin cytoskeleton [25]. Due to the sensitive inducibility of cytosolic free [Ca^2+^] by multiple environmental stresses, it is obvious that the CAM/CML family should be coordinated with responses to biotic and abiotic stresses, such as the roles of AtCML43 and pepper CaCAM1 in resistance to pathogens [26,27], AtCML9 in tolerance to drought and salinity stress [21], AtCAM3 in thermotolerance [28], sweet potato SpCAM in salt-stress-mediated leaf senescence [29], *Glycine soja* GsCML27 in bicarbonate stress tolerance and negative regulation of salt stress or osmotic stress [30], and rice OsCML16 in drought resistance [31].

Compared to various studies on members of the *CAM/CML* families from angiosperm lineages, much less investigation has been devoted to those from gymnosperms. With the recent demonstration of calcium playing a role in the early development of *Ginkgo biloba* L. ovules—i.e., the specific formation of pollen chambers [32]—it is feasible that the calcium signaling mediated by CAMs/CMLs could participate in developmental regulation in such an ancient relict plant. The main objectives of this study were to characterize *CAMs/CML*s in *G. biloba*, together with their spatial and temporal expression profiles, and comparatively assess their conservation and evolutionary relationships with those in *Arabidopsis*. Moreover, a representative *GbCML* (*Gb_30819*) was investigated for its function, through both ectopic expression within the transgenic *Arabidopsis* and assay of Ca^2+^-binding activity in vitro. The present study also provides evidence for extensive conservation of calcium signaling throughout the seed-bearing plants.

## 2. Results

### 2.1. Identification of CAM/CML Homologs

A total of 26 *CAM/CML* genes (designated as *GbCAMs/GbCML*s) were retrieved from the genome of *G. biloba* (Appendix A). These GbCAMs/GbCMLs consisted of amino acids from 149 (Gb_08148 and Gb_13552) to 286 (Gb_11458), with molecular mass from 16.86 (Gb_13552) to 32.39 kDa (Gb_11458), and pI from 3.89 (Gb_08148 and Gb_13552) to 5.87 (Gb_22573). As the comparative orthologs, seven *AtCAM*s and 50 *AtCML*s were identified from the model plant *Arabidopsis*. Correspondingly, they had amino acid numbers from 83 (AT3G03430 and AT5G17480) to 354 (AT5G04170), molecular mass from 9.05 (AT5G17480) to 37.11 kDa (AT5G04170), and pI from 3.61 (AT3G22930) to 7.76 (AT3G10300).

The identified *CAM/CML* homologs were unevenly distributed in the individual genomes (Appendix A, Figure 1). For instance, *GbCAMs/GbCML*s were mapped among 11 out of 12 chromosomes in the *Ginkgo* genome, except for chromosome no.6 (Figure 1a). Among these chromosomes, No. 4, No. 10, and No. 12 had the highest numbers (four). Each of four *Ginkgo* chromosomes (No. 1, No. 5, No. 8, and No. 11) contained one *GbCAM/GbCML*, whereas both chromosomes no.2 and no.7 had two members, with three *GbCAMs/GbCML*s on the remaining chromosomes (no.3 and no.9). *AtCAMs/AtCML*s were distributed with the maximum (20 members) and the minimum (6 members) on *Arabidopsis* chromosomes no.3 and no.4, respectively (Figure 1b), and both chromosomes no.2 and no.5 had 9 *AtCAMs/AtCML*s, in contrast to chromosome no.1 with 13 members (Figure 1b).

### 2.2. Phylogenetic and Gene Structural Analysis of CAMs/CMLs

The constructed phylogenetic tree showed that these CAMs/CMLs (totaling 83 members) could be clustered into 10 subgroups, among which the highest numbers were 18 in subgroup I, followed by 12 in subgroup VIII (Figure 2), while subgroups II and X had the least members (2 and 3, respectively), all of which were from AtCMLs. As shown in Figure 2, the *Ginkgo* CAM/CML proteins were grouped into different subgroups rather than a single one. Accordingly, out of 26 GbCAMs/GbCMLs, 7 members were in subgroup IV, 6 in subgroup I, 5 in subgroup VIII, 3 in subgroup III, 2 in subgroups V and IX, and 1 in subgroup VII (Figure 2). The distribution of 57 AtCAMs/AtCMLs was also exhibited in a similar pattern, with varied numbers included in different subgroups. However, all seven AtCAMs (i.e., AT1G66410, AT2G27030, AT2G41110, AT3G43810, AT3G56800, AT5G21274, and AT5G37780) were clustered into subgroup VIII, together with five GbCAMs/GbCMLs (Figure 2).

The phylogenetic ML tree was constructed based on amino acid sequences (Figure 2); therefore, the sequence identity among these identified CAMs/CMLs was further compared (Appendix A). It was found that the sequence identity among all 83 CAMs/CMLs was 18.15%, whereas those among the 50 AtCMLs and 26 GbCAMs/GbCMLs were 18.57% or 23.49% on average, respectively. Comparatively, seven AtCAMs presented significantly higher sequence identity, with a value of 98.85% on average. Furthermore, when compared with a typical CAM (i.e., AtCAM2, with gene ID AT2G41110), five GbCAMs/GbCMLs—including Gb_13552 (93.96%), Gb_08148 (93.29%), Gb_17936 (67.74%), Gb_30717 (65.44%), and Gb_20553 (51.02%)—displayed more than 50% sequence identity (Appendix A). Moreover, they were clustered into subgroup VIII together with 7sevenAtCAMs in the phylogenetic tree (Figure 2). Therefore, they could be classified as CAMs in *G. biloba*, i.e., GbCAMs. As a result, the identified GbCAMs/GbCMLs should be composed of 5 GbCAMs and 21GbCMLs. The 21 GbCMLs had lower identity, with divergent values ranging from 18.62% (Gb_11458) to 42.08% (Gb_30819), compared with the *Arabidopsis* CML AtCML5 (AT2G43290, Appendix A).

The existence of a certain degree of multiplicity of GbCAMs/GbCMLs was also revealed with respect to the composition of the EF-hand domains. Most of them (17 out of 26) had two pairs of EF hands, with the remaining members having at least one pair of EF hands (Figure 2, Figure 3 and Figure 4). Varying numbers of EF hands were also observed within the 57 AtCAMs/AtCMLs (Figure 2, Figure 3 and Figure 4). Together with 7 AtCAMs, 31 out of 50 AtCMLs contained two pairs of EF hands, and another 17 AtCMLs contained pair of EF hands, compared to one member (AtCML12, gene ID AT2G41100) with three pairs of EF hands. The sole exception was AtCML1 (AT3G59450), which contained no detectable EF hands (Figure 2, Figure 3 and Figure 4).

Based on their amino acid sequences, both the first and the second pairs of EF hands from individual *Ginkgo* and *Arabidopsis* CAMs/CMLs were aligned (Figure 3 and Figure 4). It was found that the amino acids responsible for Ca^2+^ binding within the loop structure of individual EF hands were highly conserved, such as the 1st (aspartate D), 3rd (D or asparagine N), 5th (D or N), and 12th (glutamate E) amino acids. Additional conservation within the loop structure was represented by two amino acids at the 6th (glycine G) and 8th (isoleucine I) positions, which were predicted to play roles in the loop stability (Figure 3 and Figure 4). However, certain variants of amino acids were present within the loop of the #1 EF hand, such as position 1 of L (leucine) from Gb_05638, Gb_05639, and Gb_34519, or M (methionine) from Gb_28442, instead of D; and position 12 of D within the loops of the #1 EF hand (Gb_05638 and Gb_05639), the #2 EF hand (Gb_13855, Gb_13856, and Gb_22573), and the #3 EF hand (Gb_11457, Gb_11458, and Gb_15095), rather than E (Figure 3 and Figure 4). Similarly, members of the 50 AtCMLs also contained some variations in their EF hands (Figure 3 and Figure 4).

To characterize their gene structural diversity, the exon–intron patterns of the 83 *CAMs/CML*s were analyzed (Figure 5a). The results revealed that 60 out of 83 *CAMs/CML*s contained a single exon without introns, including the majority (21/26) of the *GbCAMs/GbCML*s. These intronless *CAMs/CML*s were generally classified into subgroups I to VI, with exception of three genes (*AT3G24110*, *AT4G26470*, and *AT3G59450*) that possessed four exons. In contrast, those *CAMs/CML*s with a number of exons > 1 were composed of 2–6 exons per gene, and distributed into subgroups VII to X, respectively. Two *AtCAM*s (*AT4G37010* and *AT3G50360*) had the maximum number of six exons. Generally, the *CAM/CML* gene structures of a specific clade were similar to one another (Figure 5a). This similarity was also demonstrated by MEME analysis (Figure 5b). Among 10 of the conserved motifs of CAM/CML proteins, the motif combinations—i.e., motif01 + motif02/07 + motif01—representing individual EF hands appeared in most of the clades (Figure 5b).

### 2.3. Collinearity Analysis of CAMs/CMLs

Using the software MCScanX, a total of 12 *CAM/CML* gene pairs with collinearity relationships were identified, all of which were composed of the intraspecies pairs from *Arabidopsis* (Figure 6a), without any intraspecies pairs from *Ginkgo* (Figure 6b) or interspecies pairs across *Arabidopsis* and *Ginkgo* (Figure 6c), indicating no synteny relationships among the *GbCAMs/GbCML*s, or between the *GbCAMs/GbCML*s and *AtCAMs/AtCML*s. Subsequently, the tandem duplicated genes were screened against each genome of *Arabidopsis* and *Ginkgo*. As a result, three gene pairs ascribed to tandem repeats within 26 *GbCAMs/GbCML*s (i.e., *Gb_05638*-*Gb_05639*, *Gb_13855*-*Gb_13856*, and *Gb_11457*-*Gb_11458*) and 57 *AtCAMs/AtCML*s (i.e., *AT1G76640*-*AT1G76650*, *AT2G41100*-*AT2G41110*, and *AT3G59440*-*AT3G59450*) were identified (Figure 6a,b).

### 2.4. Quantitative Analysis of GbCAM/GbCML Expression in G. biloba

To unravel their transcriptional profiles in both vegetative tissues and ovules at the early developmental stages in *G. biloba*, all 26 *GbCAMs/GbCML*s were analyzed (Figure 7). As visualized via heatmap plotting in Figure 7, their expression patterns were grouped into with three types: (I) higher, (II) lower, and (III) moderate expression. Overall, 5 out of 26 *GbCAMs/GbCML*s (i.e., *Gb_03898*, *Gb_08148*, *Gb_13552*, *Gb_30819*, and *Gb_35180*) were characterized as having higher expression levels (pattern I), in contrast to 9 (i.e., *Gb_09531*, *Gb_11457*, *Gb_11458*, *Gb_13868*, *Gb_15575*, *Gb_15581*, *Gb_30178*, *Gb_34519*, and *Gb_37768*) presenting with extremely low or no detectable expression at the transcriptional level (pattern II). Pattern III consisted of the remaining 12 *GbCAMs/GbCMLs*, including *Gb_05638*, *Gb_05639*, *Gb_09202*, *Gb_13855*, *Gb_13856*, *Gb_15095*, *Gb_16484*, *Gb_17936*, *Gb_20553*, *Gb_22573*, *Gb_28442*, and *Gb_30717*, whose expression levels were between those of patterns I and II.

With regard to the tissue-specific expression of the 26 *GbCAMs/GbCML*s, it was found that all of them were constitutively expressed in the vegetative tissues tested, albeit with differential expression levels (no significance for each gene, *p* > 0.05) between the roots, stems, and leaves. In the tissues of roots and leaves, the gene *Gb_03898* showed maximum relative expression of 12.38 and 10.51, respectively, while in stems, *Gb_35180* presented the highest value, at 10.98. On the other hand, throughout the four developmental stages of the ovules, expression levels of five *GbCML*s from pattern II (i.e., *Gb_09531*, *Gb_13868*, *Gb_15581*, *Gb_30178*, and *Gb_37768*) were not detectable (Figure 7). Within ovules, the *GbCAMs/GbCML*s from patterns I and III showed apparent differences in expression levels at the four developmental stages. A gene from pattern I (*Gb_30819*) was significantly differentially expressed (*p* < 0.05) in a trend of gradual elevation from stages I to III, then decreased at stage IV (Figure 7). Moreover, the expression levels of *Gb_30819* appeared higher than those of other *GbCML*s at the four developmental stages of early ovules, with peak expression of 43.02 at stage III. Interestingly, five *GbCAM*s (i.e., *Gb_08148*, *Gb_13552*, *Gb_17936*, *Gb_20553*, and *Gb_30717*), together with seven *AtCAM*s within subgroup VIII of the phylogenetic tree (Figure 2), had constitutive expression patterns in both vegetative tissues and ovules of *G. biloba* (Figure 7). It was notable that each of six paralogous gene pairs (Figure 2)—including *Gb_03898*-*Gb_09531*, *Gb_05638*-*Gb_05639*, *Gb_08148*-*Gb_13552*, *Gb_13855*-*Gb_13856*, *Gb_15575*-*Gb_15581*, and *Gb_17936*-*Gb_20553*—displayed differential expression, with higher magnitude in one than in its paralogous gene (Figure 7).

### 2.5. Analysis of Cis-Acting Regulatory Elements in GbCAM/GbCML Promotors

Due to apparent divergence in the expression levels of *GbCAMs/GbCML*s in various tissues and ovule developmental stages, the promotors of individual *GbCAMs/GbCML*s were predicted for cis-acting regulatory elements. The results demonstrated two main classifications of regulatory elements with varied counts in the 26 *GbCAM/GbCML* promotors: one was involved in phytohormone-responsiveness, including abscisic acid, ethylene, gibberellin, MeJA, and salicylic acid; the other was associated with the responsiveness to environmental factors—such as light, defense, or stress—and transcription factors such as Myb or Myc binding (Figure 8). Two of these cis-acting regulatory elements—i.e., Myb or Myc binding, and light responsiveness—were mapped within all 26 *GbCAM/GbCML* promotors, followed by abscisic acid responsiveness (22 promotors). The minimal mapping element was that of defense or stress responsiveness, found within nine promotors. The promotor of *Gb_30819* contained seven types of the predicted cis-acting regulatory elements, with the exception of gibberellin responsiveness, whereas those of other *GbCAMs/GbCML*s lacked several elements, varying from two to four (Figure 8).

### 2.6. Transgenic Arabidopsis Plants Overexpressing Gb_30819

Based on the expression patterns of individual *GbCAMs/GbCML*s, *Gb_30819* was selected for further investigation of its functions in plant growth and development. Following the gene cloning and construction of the expression vectors pCAMBIA1301a-*Gb_30819*, the target gene was transformed into the wild-type *Arabidopsis* (WT). The T3 generation of transgenic plants with homozygous *Gb_30819* was examined for ectopic expression using qRT-PCR. It was found that, driven by the CaMV35S promoter, the relative transcriptional magnitude of *Gb_30819* within four lines (A1, A2, D1, and D2) of transgenic *Arabidopsis* seedlings varied from 0.27 (line D1) to 1.63 (line A1) (Figure 9a). Thereafter, due to the overexpression of *Gb_30819*, the transgenic *Arabidopsis* line A1 was prepared for analysis of plant phenotypes.

The *Gb_30819* transgenic plants (T3 generation) exhibited several aspects of variant phenotypes at both the seedling and fruiting periods (Figure 9b–e). As shown in Figure 9b,c, the transgenic seedlings saw a significant shortening in root length (i.e., 0.75 cm on average) and height (0.58 cm on average), compared to WT plants, with root length of 3.29 cm on average and height of 1.05 cm on average. When these plants were grown into the fruiting period, it was observed that the yields of both pods and seeds were significantly decreased within the transgenic plants, containing 11.18 pods per plant and 26.78 seeds per pod, in contrast to 16.81 pods per plant and 37.68 seeds per pod within WTs (Figure 9d,e).

### 2.7. In Vitro Ca^2+^-Binding Activity of Gb_30819 Determined by Electrophoretic Mobility Shift Assay

After purification through glutathione–Sepharose 4B beads with affinity for GST-tagged proteins, the fusion protein GST-Gb_30819 was assayed for its Ca^2+^-binding activity in vitro. The predicted molecular weight of GST-Gb_30819 was approximately 51.2 kDa, composed of a 25.0 kDa GST and a 26.2 kDa Gb_30819. When the fusion protein GST-Gb_30819 was run in SDS–PAGE, a band corresponding to its molecular weight (51.2 kDa) was detected (Figure 10, lane 1). Comparatively, the fusion protein GST-Gb_30819 in a status of Ca^2+^ binding showed a shift in its electrophoretic mobility, with a band of faster movement (Figure 10, lane 3) than those from GST-Gb_30819 alone (Figure 10, lane 1) or the application of EDTA to GST-Gb_30819 (Figure 10, lane 2).

## 3. Discussion

CAMs/CMLs play essential roles in modulating a variety of developmental processes and stress responses, by mediating Ca^2+^ signatures [1,3,4]. Although CAMs are widely distributed in animals and plants, their homologous proteins (CMLs) are specifically presented in plants [3,8,9]. In this study, a total of 83 candidate CAMs/CMLs was identified, all of which were validated with only one type of conserved domain, i.e., EF hand (Figure 2, Figure 3 and Figure 4). Among these identified CAMs/CMLs, the numbers of EF hands within 7 AtCAMs and 50 AtCMLs in *Arabidopsis* were consistent with those in previous reports [8,10]. One of the *Arabidopsis* AtCMLs (AtCML1, with gene ID AT3G59450) had no identifiable EF hand (Figure 3 and Figure 4). However, it was retained in this study for taking into consideration consistency with previous research on *Arabidopsis* CAMs/CMLs [10]. The identified GbCAMs/GbCMLs in *G. biloba* displayed multiplicity at several layers. Firstly, the comparison of amino acid sequences between individual GbCAMs and AtCAM2 (AT2G41110, Appendix A), showed all five GbCAMs with sequence identities > 50%, including Gb_13552 (93.96%), Gb_08148 (93.29%), Gb_17936 (67.74%), Gb_30717 (65.44%), and Gb_20553 (51.02%), which could be classified as CAM proteins based on other studies [8]. Meanwhile, the 21 GbCMLs were varied, with sequence identities from 18.62% (Gb_11458) to 42.08% (Gb_30819), compared with AtCML5 (AT2G43290). Secondly, 17 of the GbCAMs/GbCMLs harbored two pairs of EF hands, while another 9 GbCAMs/GbCMLs had one pair of EF hands (Figure 2, Figure 3 and Figure 4). Finally, although the strong conservation of amino acids within the Ca^2+^-binding loops was observed from the alignment of the EF-hand domains among 26 GbCAMs/GbCMLs, 7 AtCAMs, and 50 AtCMLs, some variants of amino acids occurred at key positions in the loops (Figure 3 and Figure 4). Variations of GbCAMs/GbCMLs—in terms of their composition of EF hands and the conserved amino acids for Ca^2+^-chelating in the loops—were likely to result in distinct Ca^2+^-binding activities [7,9].

As shown in Figure 1 and Figure 6, members of the *CAM/CML* gene family were characterized by uneven chromosomal localization. Maximal numbers of 20 and 4 were localized on the chromosomes of *Arabidopsis* and *Ginkgo*, respectively. Accumulated evidence suggests that *CAM/CML* families in various species are composed of a multitude of members—for instance, 5 *CAM*s and 32 *CML*s in rice [11], 4 *CAM*s and 36 *CML*s in *F. vesca* [12], 52 *CML*s in *S. lycopersicum* [13], 79 *CML*s in *B. rapa* [14], 3 *CAM*s and 62 *CML*s in *V. vinifera* [15], and 4 *CAM*s and 58 *CML*s in *M. domestica* [16]. The formation of multigene families has been associated with gene duplications, which are likely driven by tandem, segmental duplication or whole-genome duplication, as well as transposition duplication [33]. According to the comparative characterization of CAM/CML evolution in the green lineage, Zhu et al. proposed that expansion of the CAM/CML family in plants was associated with efficient processing of environmental signals and promotion of adaptation to land environments [17]. To unravel the mechanisms of duplication of the CAM/CML family of both *Arabidopsis* and *Ginkgo*, collinearity relationships among these members were investigated. Collinearity analysis showed that no synteny relationships were present among the *GbCAMs/GbCML*s, or between the *GbCAMs/GbCML*s and *AtCAMs/AtCML*s (Figure 6b,c), suggesting that segmental or whole-genome duplications might not have contributed to the expansion of the *CAM/CML* gene family in *G. biloba*. On the other hand, 12 gene pairs with collinearity relationships were demonstrated among the *AtCAMs/AtCML*s (Figure 6a). When analyzing tandem duplication, three tandem-duplicated gene pairs were found from *Ginkgo* and *Arabidopsis CAM/CML* members (Figure 6a,b). Therefore, it was inferred that tandem duplications were likely to play roles in the evolution of the *Ginkgo CAM/CML* family, in contrast to both tandem and segmental or whole-genome duplications in that of *Arabidopsis*.

Within the constructed phylogenetic tree (Figure 2), the identified CAMs/CMLs (totaling 83 members) were dispersed into 10 subgroups, among which 57 AtCAMs/AtCMLs were similarly clustered with the previous tree constructed using these AtCAMs/AtCMLs alone [10]. The results indicated that the degrees of sequence identity were consistent with their phylogenetic relationships (Appendix A). The phylogenetic relationships of 83 CAMs/CMLs were also coordinated with their exon–intron and MEME patterns (Figure 5). It is noteworthy that five GbCAMs were clustered into one subgroup (VIII), together with seven AtCAMs (Figure 2), supporting their close phylogenetic relationship and high sequence identity. Moreover, these GbCAMs (i.e., *Gb_08148*, *Gb_13552*, *Gb_17936*, *Gb_20553*, and *Gb_30717*) were constitutively expressed throughout the vegetative tissues and ovules at the early developmental stages (Figure 7). The property of constitutive expression has also been reported for seven AtCAMs [10]. The results suggest that CAMs might function as housekeeping genes involved in various physiological processes.

The identified GbCMLs (21 members) were clustered into subgroups I (6 GbCMLs), III (3), IV (7), V (2), VII (1), and IX (2) (Figure 2). As a result, six paralogous gene pairs—including *Gb_03898*-*Gb_09531*, *Gb_05638*-*Gb_05639*, *Gb_08148*-*Gb_13552*, *Gb_13855*-*Gb_13856*, *Gb_15575*-*Gb_15581*, and *Gb_17936*-*Gb_20553*—were observed (Figure 2). In many cases, the paralogous gene pair presented higher expression levels in one gene than the other (Figure 7). Based on their relative expression levels, 26 *GbCAMs/GbCML*s were grouped into three types of patterns: (I) higher, (II) lower, and (III) moderate. One explanation for the varied expression profiles of *GbCAMs/GbCML*s is the specific cis-acting regulatory elements in their promotors. Analysis of the promotors of *GbCAMs/GbCML*s (Figure 8) revealed that modulation by phytohormones—such as abscisic acid, ethylene, gibberellin, MeJA, and salicylic acid—was likely to account for the temporal and spatial expression of *GbCAMs/GbCML* genes. In addition, other elements associated with the responsiveness to light, defense, or stress, or to transcription factors such as Myb or Myc binding, may be responsible for the expression features of *GbCAMs/GbCML*s exposed to diverse environmental factors, which needs to be experimentally demonstrated in future.

Based on the expression profiles of 26 *GbCAMs/GbCML*s (Figure 7), a *GbCML* gene (*Gb_30819*) showed a distinct pattern with significantly differential expression (*p* < 0.05) between the developmental stages of ovules, in addition to its higher expression levels than those of other *GbCML*s. Additionally, the promotor of *Gb_30819* contained multiple cis-acting regulatory elements involved in phytohormone-responsiveness, such as abscisic acid, ethylene, MeJA, and salicylic acid (Figure 8). Therefore, the *Gb_30819* gene was chosen for the analysis of its functions. The ectopic expression of the gene *Gb_30819* in transgenic *Arabidopsis* conferred a dwarf phenotype to their seedlings, characterized by significantly shortened root length and seedling height compared to those of wild-type plants (Figure 9b,c). Furthermore, another divergent phenotype with significantly decreased yields of both pods and seeds was observed in the fruiting period of the Gb_30819 transgenic plants (Figure 9d,e). Investigation of the functions of other *CAM/CML* genes via ectopic expression has also been reported. For instance, transgenic plants overexpressing *AtCAM7* displayed positive effects on photomorphogenic growth [18], while *AtCML43* and pepper *CaCAM1* overexpressed in transgenic plants could promote pathogen resistance [26,27]. An *Arabidopsis* CML—AT2G43290 (AtCML5) [10]—is the closest homolog for Gb_30819, with an amino acid sequence identity of 42.08% (Appendix A). Previous research has revealed that the *AtCML5* gene is varied in its expression levels throughout the full developmental stages of *Arabidopsis*, with the maximal amount at the late flowering stage [10]. Analogously, *Gb_30819* was a transcriptionally active gene with higher expression (Figure 7). Although it was constitutively expressed in both vegetative tissues and ovules in *G. biloba*, *Gb_30819* appeared to show temporal expression with differential levels among the four developmental stages of ovules, with the peak value at stage III (Figure 7). Moreover, the ability of Gb_30819 to interact with Ca^2+^—as demonstrated from its strong conservation of amino acids within the Ca^2+^-binding loops of EF-hand domains (Figure 3 and Figure 4) and the electrophoretic mobility shift assay (Figure 10)—suggested that the phenotype in *Gb_30819* transgenic plants might be concerned with the modulation of Ca^2+^ signaling. The research by Ruge et al. has proven that AtCML5, functioning as a cytosol Ca^2+^ sensor, could play another role in transporting vesicles between various endomembrane structures [34]. Although it showed relatively closer homology to AtCML5 than other GbCMLs did, the CML protein Gb_30819 in *G. biloba* showed a great sequence divergence, with 42.08% identity between them. Thus, further investigations are needed to unravel the functional mechanisms of Gb_30819.

## 4. Materials and Methods

### 4.1. CAM/CML Gene Identification

The genome data were downloaded from the databases Ensembl Genomes (http://ensemblgenomes.org (accessed on 4 October 2020)) for *A. thaliana* and *GigaDB* (http://gigadb.org/dataset (accessed on 10 September 2020)) for *G. biloba*. To identify *CAM/CML* gene family members, the specific hidden Markov model (HMM)—i.e., EF-hand_7.hmm (PF13499)—was used as a query to retrieve the individual data using the program BLASTP (E-value < 1 × 10^−5^). The primary sequences were subsequently validated by removing those EF-hand-domain-containing proteins with other functional HMM motifs, while the remaining sequences containing only EF-hand domains were reserved as the potential CAM/CML members [35].

### 4.2. Phylogenetic Tree Construction

The amino acid sequences were aligned using the program MUSCLE [36]. The files from the multiple sequence alignment were input to the program FastTree with a model of JJT and a bootstrap value of 1000 to construct a maximum likelihood (ML) tree [37]. Visualization of the constructed phylogenetic tree was carried out via the web server iTOL (https://itol.embl.de (accessed on 12 December 2021)).

### 4.3. Molecular Features, Gene Structures, and Conserved Motifs of CAM/CML Proteins

The molecular features—including amino acid length of peptides, protein weight, and isoelectric point (pI)—were obtained by online searching via the tools of ExPASy ProtParam (https://web.expasy.org/protparam, accessed on 20 October 2021). After retrieving the information of genes’ structure against their genomic data, the intron–exon patterns of each *CAM/CML* gene were displayed through the GSDS server (http://gsds.gao-lab.org, accessed on 16 October 2021). To exhibit the degree of conservation, MEME analysis of the identified CAM/CML proteins was performed under the setting of 10 motifs and other default parameters (https://meme-suite.org/meme/tools/meme (accessed on 17 October 2021)).

### 4.4. Chromosomal Distributions and Collinearity Relationships of CAM/CML Genes

Chromosomal distributions of individual *CAM/CML* genes were based on their genomic annotation files. Intraspecies or interspecies collinearity relationships between *CAM/CML*s were searched using MCScanX [38]. The methods of data preparation and execution of blasting were as described previously [39].

### 4.5. Expression Analysis of GbCAMs/GbCMLs in G. biloba

For qRT-PCR (quantitative real-time PCR) analysis of *GbCAM/GbCML* expression in the vegetative tissues (i.e., roots, stems, and leaves), four-year-old seedlings of *G. biloba* were collected to provide corresponding samples. The ovules were sampled from trees of >20-year-old female *Ginkgo*. To quantify the expression levels of *GbCAMs/GbCML*s in ovules, sampling of ovules at four developmental stages was carried out according to the previous definition for specific stages of early ovules in *G. biloba* by Li et al. [32,40]. In brief, after the RNA extraction (by the RNAprep Pure Plant Kit, Tiangen Biotech, Beijing, China) and cDNA synthesis (by the Quantscript RT Kit, Tiangen Biotech), PCR was performed following the procedure of the SYBR Green qRT-PCR Kit (Tiangen Biotech), with corresponding primer pairs for *GbCAMs/GbCML*s (Appendix A). The *GAPDH* gene was used as an internal reference to calculate relative expression levels, using the 2^−∆∆Ct^ method [41]. Three parallel assays were carried out for quantitative analysis of the expression of each *GbCAM/GbCML*.

### 4.6. Analysis of Regulatory Elements in GbCAM/GbCML Promotors

Promotors were considered as 2 kb sequences in length and compiled from upstream of each *GbCAM/GbCML* gene within the Ginkgo genome data. The files of promotor sequences, in FATSA format, were uploaded onto the web server PlantCARE for online prediction (http://bioinformatics.psb.ugent.be/webtools/plantcare, accessed on 2 March 2021).

### 4.7. Construction of the Eukaryotic Expression Vector pCAMBIA1301a-Gb_30819 and Arabidopsis Transformation

Full-length cDNA of *Gb_30819* was first cloned into pMD18-T with the specific primer pairs containing the recognized sites for double-cutting by the restriction endonucleases of *BamH*I and *Hind*III (Appendix A). Subsequently, both vectors—pMD18-T- *Gb_30819* and pCAMBIA1301a—were cut with *BamH*I and *Hind*III, followed by recovery and purification of the target fragments. Finally, the purified fragments of *Gb_30819* and pCAMBIA1301a were constructed into the eukaryotic expression vector pCAMBIA1301a-*Gb_30819* using T4 DNA ligase. Within pCAMBIA1301a-*Gb_30819*, the transcriptional expression of *Gb_30819* was under the control of the constitutive promoter of CaMV35S.

Wild-type *Arabidopsis* plants (Col-0) were transformed by *Agrobacterium tumefaciens* (GV3101)/pCAMBIA1301a-*Gb_30819*, using the floral dip method. After kanamycin-resistance selection and PCR validation, the positive *Gb_30819* transgenic plants were propagated via selfing to generate homozygous lines at the third filial generation (T3). For phenotype observation in plants, seeds of wild-type or transgenic *Arabidopsis* were grown on MS plates or in pots, respectively. Two-week-old seedlings were measured for their root length and height. Fruiting plants were counted for yields of pods and seeds from the eighth week to the natural end of growth.

### 4.8. Bacterial Expression of the Fusion Protein Gb_30819, and Protein Mobility Shift Electrophoresis Assay

The *Gb_30819* cDNA was cloned into the vector pGEX4T1, producing the bacterial expression vector pGEX-*Gb_30819*, which was subsequently introduced into *E. coli* BL21. *E. coli* BL21/pGEX-*Gb_30819* was grown in LB liquid medium containing ampicillin for the selection of positive clones. The vectors extracted from the positive clones were further validated via PCR using the *Gb_30819*-specific primers. Expression of the fusion protein Gb_30819 with a tag of glutathione s-transferase (GST) was induced by isopropyl thiogalactopyranoside. Then, the bacterial cell lysate was obtained by sonication at 4 °C, with its supernatant containing the soluble Gb_30819 protein. The fusion protein Gb_30819 was purified with beads of glutathione–Sepharose 4B according to the manufacturer’s instructions (General Electric Healthcare, Chicago, IL, USA). To test the Ca^2+^-binding activity of the protein in vitro, a protein mobility shift electrophoresis assay was performed [42]. For assay of the fusion protein Gb_30819, three sets of parallel systems were prepared: (1) 3 μg of the purified Gb_30819; (2) an incubated mixture with the purified Gb_30819 and 2 mM EDTA (ethylenediaminetetraacetic acid); and (3) another with the purified Gb_30819 and 2 mM CaCl_2_. Electrophoresis was run in a 15% gel of SDS–PAGE.

### 4.9. Statistical Analysis of Data

Statistical data processing was carried out via Student’s *t*-test (two-tailed test). Statistically significant differences were considered at *p* < 0.05 or *p* < 0.01. Statistical data in figures were marked as mean values ± standard deviation (*n* = 4 or 5).

## 5. Conclusions

Compared with those from *A. thaliana*, 5 *GbCAM*s and 21 *GbCML*s from *G. biloba* displayed a certain degree of multiplicity in both their sequences and expression profiles, although they possessed highly conserved EF-hand domains, inferring that they could have undergone evolutionary diversification of functions. Collinearity analysis suggested that tandem rather than segmental or whole-genome duplications were likely to play roles in the evolution of the *Ginkgo* CAM/CML family. Furthermore, a potential role of the gene *Gb_30819* in developmental modulation was determined from the causative influence of its ectopic expression on the seedling and fruit development of transgenic *Arabidopsis*. Altogether, the present research contributes to insights into the characteristics of the evolution and expression of *GbCAMs/GbCML*s, as well as evidence for Ca^2+^-CAM/CML pathways functioning within the ancient gymnosperm *G. biloba*.

## Figures and Tables

**Figure 1 plants-11-01506-f001:**
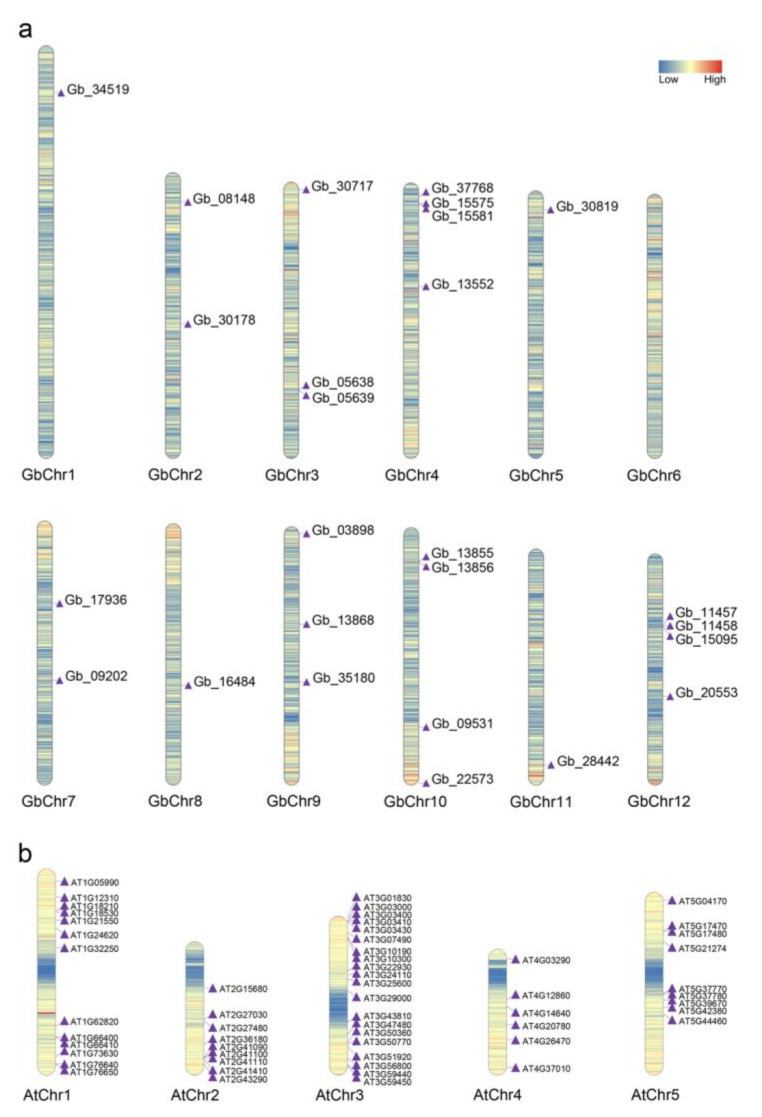
*CAMs/CML*s’ localization on individual chromosomes: (**a**) *GbCAMs/GbCML*s; (**b**) *AtCAMs/AtCML*s. Gene density of chromosomes from lower to higher is indicated from blue to red within the bar, respectively.

**Figure 2 plants-11-01506-f002:**
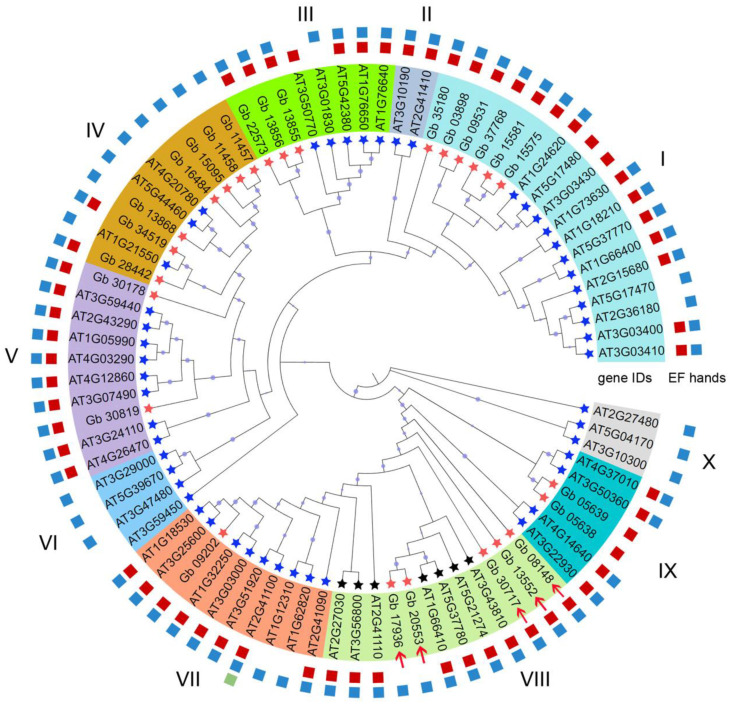
Phylogenetic tree constructed with 83 CAMs/CMLs. Ten subgroups are indicated with I–X, respectively. Squares in red, light blue, and light green refer to pairs of EF-hand domains. Arrows in red indicate the 5 GbCAMs (Gb_08148, Gb_13552, Gb_17936, Gb_20553, and Gb_30717) within subgroup VIII. Circles at the individual nodes represent bootstrap support.

**Figure 3 plants-11-01506-f003:**
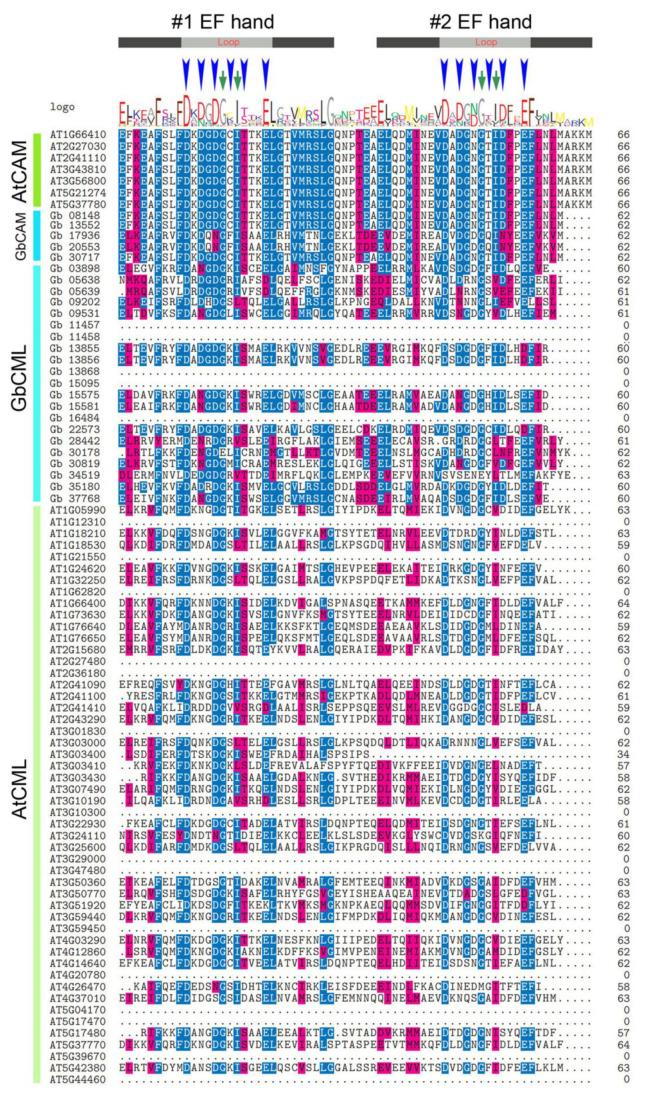
Sequence alignment of the first pair of EF-hand domains among the 83 CAM/CMLs identified. The two EF hands are marked with #1 and #2 EF hands. The absence of EF hands is represented with dots. Blue arrowheads indicate the amino acids responsible for Ca^2+^ binding within the loop structure. Green arrows indicate amino acids responsible for loop stability. Letters with different background are the conserved amino acids between these sequences.

**Figure 4 plants-11-01506-f004:**
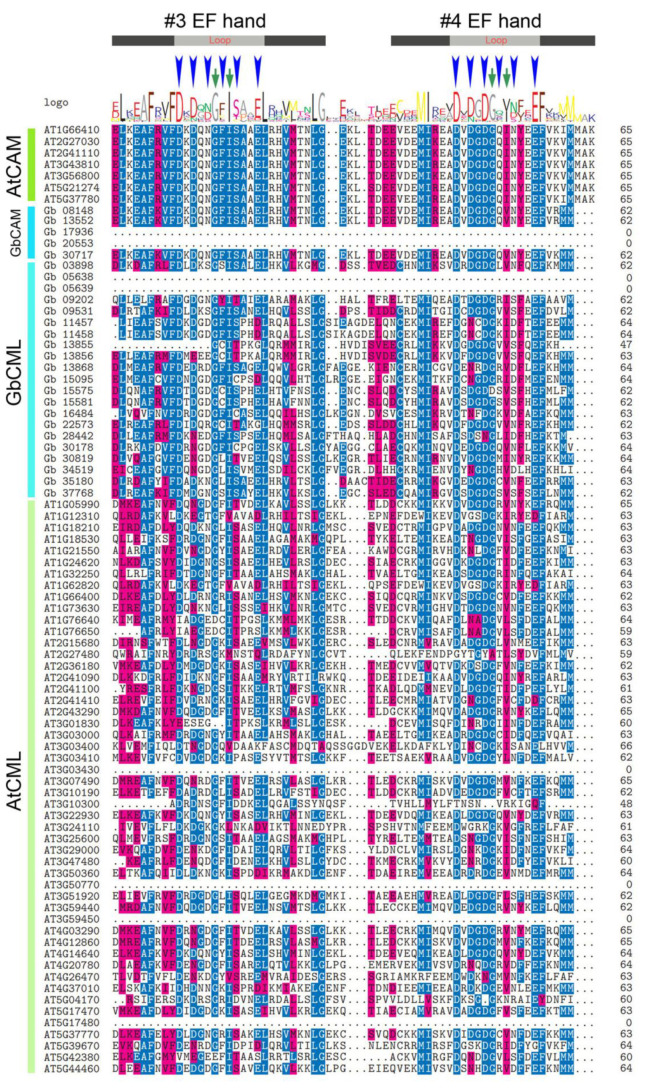
Sequence alignment of the second pair of EF-hand domains among the 83 CAM/CMLs identified. The two EF hands are marked with #3 and #4 EF hands. Symbols refer to the descriptions in Figure 3. Letters with different background are the conserved amino acids between these sequences.

**Figure 5 plants-11-01506-f005:**
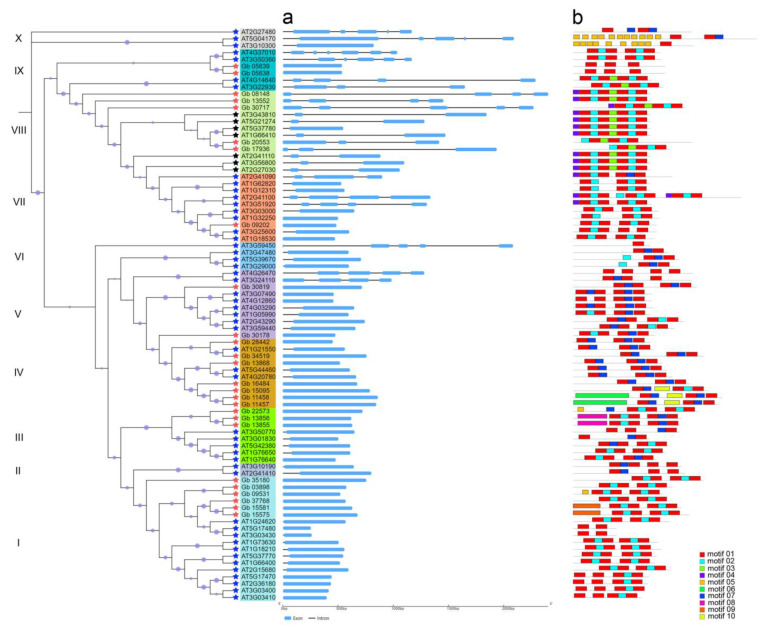
(**a**) Exon–intron and (**b**) MEME patterns of 83 *CAMs/CML*s.

**Figure 6 plants-11-01506-f006:**
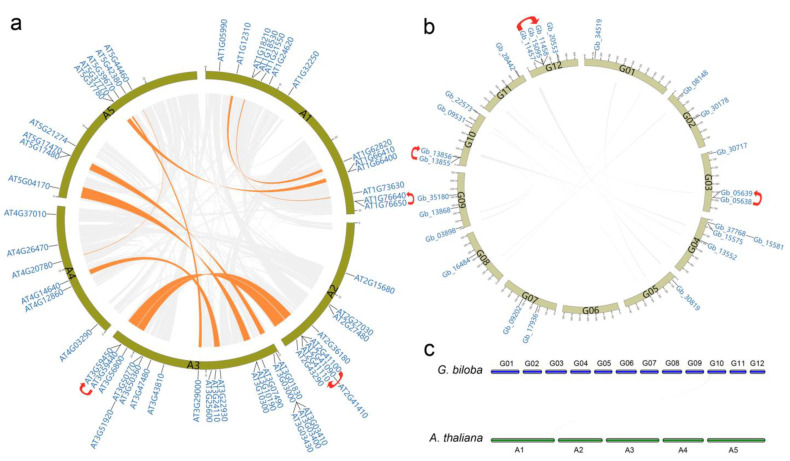
Collinearity relationships between (**a**) *AtCAMs/AtCML*s, (**b)**
*GbCAMs/GbCML*s, and (**c)**
*GbCAMs/GbCML*s and *AtCAMs/AtCML*s. A1–A5 and G01–G12 refer to chromosomes of *Arabidopsis* and *Ginkgo*, respectively. Linkages in orange indicate the intraspecies synteny blocks, containing the *CAM/CML* gene pairs with collinearity, while those in grey mark the blocks between genomes. Curved arrows in red mark gene pairs of tandem duplication. Gene IDs labeled on chromosomes are *CAMs/CML*s identified in *Arabidopsis* and *Ginkgo*.

**Figure 7 plants-11-01506-f007:**
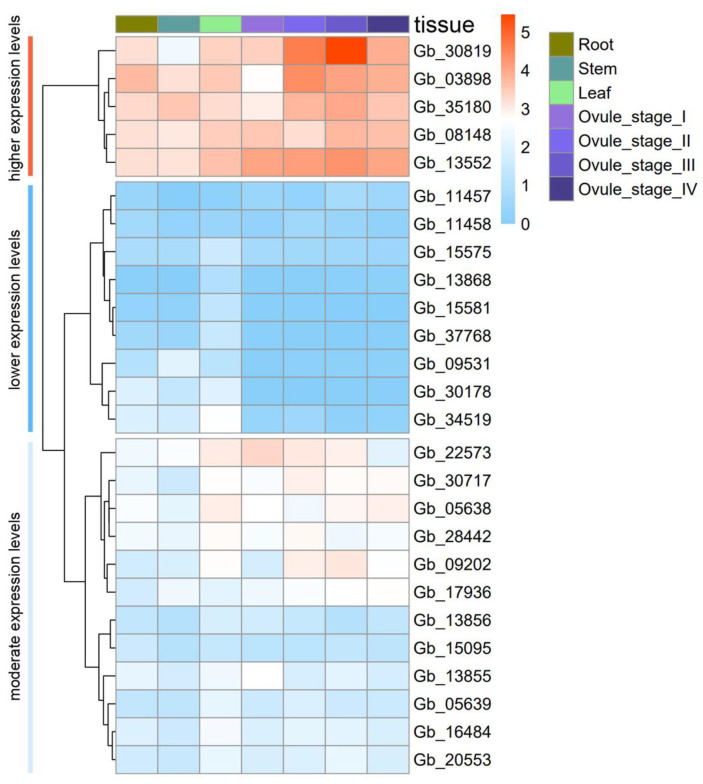
Heatmap of expression profiles of 26 *GbCAMs/GbCML*s in roots, stems, leaves, and ovules at the four developmental stages (I–IV) of *G. biloba*. The color bar from light blue to red indicates relative expression levels from lower to higher, respectively.

**Figure 8 plants-11-01506-f008:**
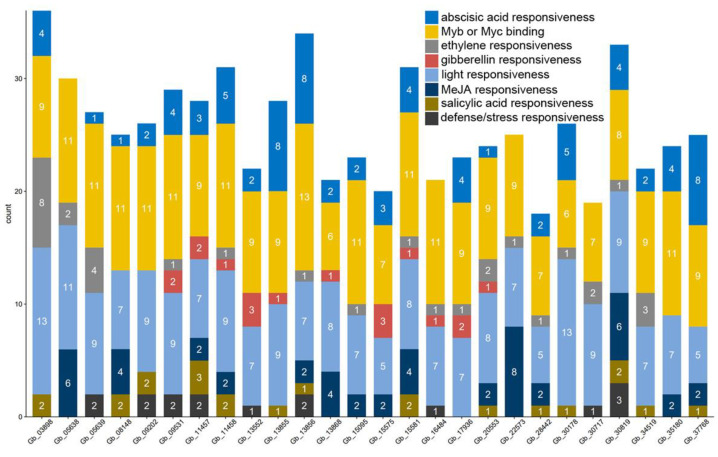
Representatives of cis-regulatory elements identified from *GbCAM/GbCML* promotors. Columns in colors refer to these elements, while numbers on columns indicate the amounts of individual elements.

**Figure 9 plants-11-01506-f009:**
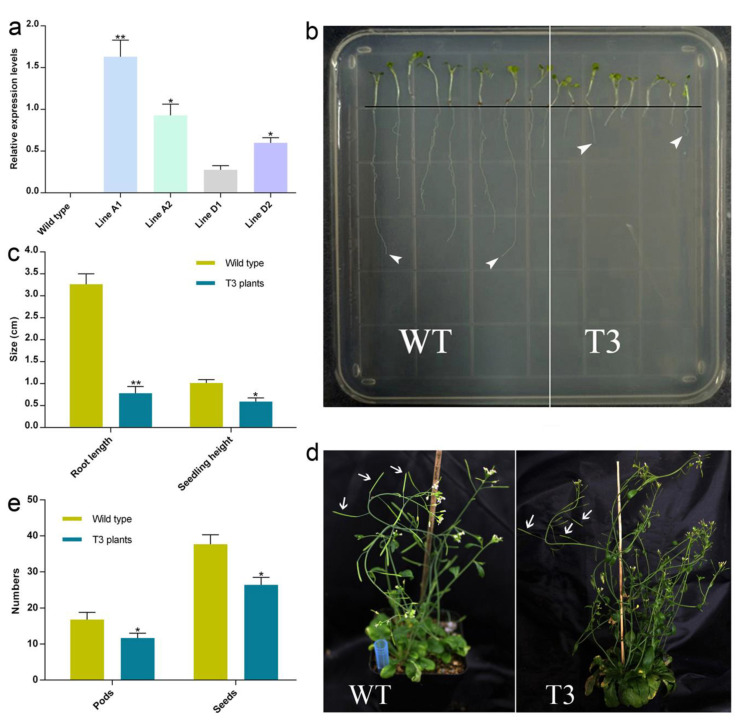
Transgenic *Arabidopsis* plants overexpressing *Gb_30819*: (**a**) Relative transcriptional magnitude of *Gb_30819* within four lines of transgenic seedlings (Lines A1, A2, D1, and D2); Wild type indicates wild-type *Arabidopsis* seedlings. (**b**) Phenotypes and (**c**) statistical plotting of root length and seedling height in transgenic (T3) and wild-type (WT) *Arabidopsis*; white arrowheads indicate root tips. (**d**) Phenotypes and (**e**) statistical plotting of pod and seed numbers in transgenic (T3) and wild-type (WT) *Arabidopsis* in the fruiting periods; white arrows indicate pods. * *p* < 0.05 and ** *p* < 0.01, Student’s *t*-test, compared with Line D1 in panel (**a**) and WT in panels (**c**,**e**).

**Figure 10 plants-11-01506-f010:**
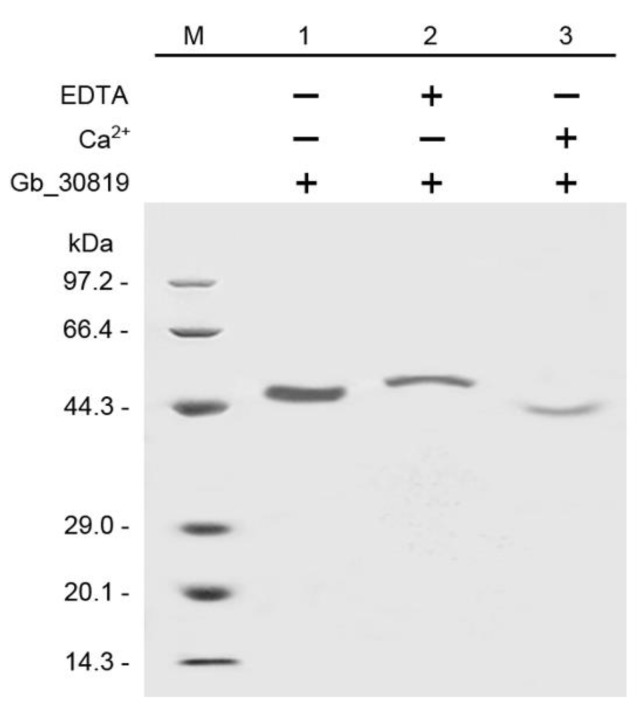
Electrophoretic mobility shift assay in SDS–PAGE. Line M, molecular mass markers of protein; Line 1, fusion protein Gb_30819; Line 2, fusion protein Gb_30819 + EDTA; Line 3, fusion protein Gb_30819 + Ca^2+^. Symbols + and – refer to with and without the corresponding components in each reaction system, respectively.

## Data Availability

The genome data of *G. biloba* were downloaded from the database GigaDB (http://gigadb.org/dataset, accessed on 10 September 2020), while those of *A. thaliana* were downloaded from Ensembl Genomes (http://ensemblgenomes.org, accessed on 4 October 2020).

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
