# Peer review of "Characterization of the Calmodulin/Calmodulin-like Protein (CAM/CML) Family in Ginkgo biloba, and the Influence of an Ectopically Expressed GbCML Gene (Gb_30819) on Seedling and Fruit Development of Transgenic Arabidopsis"

_plants, 2022, doi:10.3390/plants11111506_

Round 1
Reviewer 1 Report
This article is devoted to the study of the multigenic family of Ca-binding proteins (CAM/CML) in Ginko biloba, as well as the study of the functions of one of the representatives of this family, namely the GbCML gene (Gb_30819).
The authors have done a lot of experimental and analytical work. Modern methods of bioinformatic analysis were used in this study. The manuscript is well written and designed. However, before this manuscript can be published, some improvements should be done:
- Major Revisions
1) In the results, I did not see a clear separation of which G. biloba genes belong to the CAM family and which to the CML. Also, in my opinion, it is incorrect to compare the CAM sequences of G. biloba with the CML of Arabidopsis. The authors should indicate which G. biloba genes belong to CAM and which to CML. And also compare the CAM of Arabidopsis with Ginko CAM and CML in accordance with belonging to a particular family.
2) The chapter materials and methods does not contain the chapter statistical data processing. Also, in Figure 9a, c, e, it is necessary to indicate the veracity (p).
3) The authors should explain the choice of the Gb_30819 gene for the analysis of its functions. Also, in the discussion, the authors should indicate which is the closest homologue for it in Arabidopsis. Authors should also add information about the known functions of this homologue.
- Minor Essential Revisions
11) line 46: CaM/CML => CAM/CML
22) line 47: and At50 CML genes => and 50 AtCML genes. Also, the names of genes should be italicized.
33) line 47: earlier in the text, the authors indicated the number of genes in numbers, on this line in words: « The Arabidopsis genome harbors 7 AtCAM and 50 AtCML genes…» and «The seven CAMs were…» I recommend the authors to choose one style, how to write the number of genes or proteins, for example, only in numbers. This remark applies to the entire text of the manuscript (lines 126-128, 130-132, 174-176 and so on).
44) line 57: 17 AtCMLs with a pair of EF handss, and one (AtCML12) with three pairs of EF hands => 17 AtCMLs with one pair of EF handss, and AtCML12 with three pairs of EF hands
55) line 85: Ginkgo biloba => Ginkgo biloba L.
66) line 292: «the transgenic plants containing 11.18 of 291 pods and 26.78 of seeds per plant…» => 26.78 of seeds per plant or pods? Or 2.678 thousand of seeds per plant? As far as I know, there are normally 20-50 seeds in one pod of Arabidopsis.
77) line 322, 468: Arabidopsis => Arabidopsis
88) materials and method: I have not found a complete description of RNA isolation and cDNA production. Also, at least two reference genes are usually used for PCR RT, and preferably three. For example, GAPDH, Actin, UBQ and UBC.
99) line 486: I have not found a complete description of the PCR validation.
I recommend this Ms for publication after major revisions.
Reviewer 2 Report
In the submitted manuscript, the authors described the characteristics of the CAM/CML family in Ginkgo biloba, together with their spatial and temporal expression profiles, and a comparative assessment of their conservation and evolutionary relationships with those in Arabidopsis. Moreover, a representative GbCML (Gb_30819) was investigated on its function, through both ectopic expression within the transgenic Arabidopsis and assay of Ca2+-binding activity in vitro. These results contributed to insights into the characteristics of evolution and expression of GbCAM/GbCMLs, and evidence for Ca2+-CAM/CML pathways functioning within the ancient gymnosperm G. biloba. The present study would provide evidence for extensive conservation of calcium signaling throughout the seed plants. It is apparent that the authors have thought through the methodology of the paper well. However, it is worth refining the conclusions. Conclusions are rather a summary of the results, not conclusions themselves, it would be worth pointing to the conclusions that come from the work.
Round 2
Reviewer 1 Report
This is a wonderful work that the authors have done at a high methodological level. The authors modified the manuscript according to my recommendations. The only exception is one remark concerning the use of reference genes in PCR RT. But this remark is insignificant compared to the huge amount of research. I think that this manuscript should be accepted for publication in Plants Basel.